# Renovation of Automation System Based on Industrial Internet of Things: A Case Study of a Sewage Treatment Plant

**DOI:** 10.3390/s20082175

**Published:** 2020-04-12

**Authors:** Wanhao Zhu, Zhidong Wang, Zifan Zhang

**Affiliations:** 1School of Electrical Engineering, Guangzhou College of South China University of Technology, Guangzhou 510800, China; Zhuwh@gcu.edu.cn (W.Z.); Zhangzif@gcu.edu.cn (Z.Z.); 2Research Center of Smart Energy Technology, School of Electric Power, South China University of Technology, Guangzhou 510640, China

**Keywords:** sewage treatment, IIoT, WeChat Applet, cloud server, wide-area monitoring and control system

## Abstract

The Industrial Internet of Things (IIoT) is of great significance to the improvement of industrial efficiency and quality, and to reduce industrial costs and resources. However, there are few openly-reported practical project applications based on the IIoT up to now. For legacy automation devices in traditional industry, it is especially challenging to realize the upgrading of industrial automation adopting the IIoT technology with less investment. Based on the practical engineering experience, this paper introduces the automation renovation of a sewage treatment plant. The legacy automation devices are upgraded by the central controller of a STM32 processor (Produced by STMicroelectronics company, located in Geneva, Switzerland), and the WeChatApplet (Developed by Tencent company, located in Shenzhen, China) is used as the extended host computer. A set of remote monitoring and control systems of sewage treatment based on the IIoT is built to realize the wide-area monitoring and control of sewage treatment. The paper describes the field hardware system, wide-area monitoring and control application program, management cloud platform and security technologies in detail. The actual operation results show that the monitoring system has the requirements of high accuracy, good real-time performance, reliable operation and low cost.

## 1. Introduction

With the development of urbanization in various countries, the pollution of water resources is becoming more and more serious. The water pollution usually refers to industrial waste-water pollution, rural aquaculture waste-water pollution and domestic sewage pollution. Industrial waste water is produced in the process of industrial production, such as paper mill waste water, dyeing and weaving plant waste water, mining and processing waste water, which has the most serious pollution to the water quality. The pollution of rural breeding sewage refers to pig farms and cattle farms. The sewage produced by them from the rural environment is very big. The domestic sewage refers to the sewage produced in the process of people’s life. Although the impact of domestic sewage is small, if it is directly discharged into the river, it will cause pollution to the living environment and drinking water of the city. The treatment of water pollution has become an important issue in the current urban recycling development, and has been paid attention to by various regions [1,2]. In recent years, more efforts have been made to control water resources, a large number of new sewage treatment plants have been built to improve water quality [3].

Improving water quality in sewage treatment plants is a systematic project, including chemical pollution treatment technology, sludge microorganism treatment technology, automatic control technology and monitoring system technology [4]. The monitoring system is an important part of the sewage treatment plant, and an efficient monitoring system is the key to ensuring the automatic operation of the plant equipment.

At present, the monitoring of sewage treatment plants is mostly based on local monitoring, and the whole monitoring system is composed of a host computer and PLC (Programmable Logic Controller). Operators need to conduct data monitoring and operation locally, and cannot remotely monitor the operation of the system [5]. The traditional monitoring system of sewage treatment plants based on local data supported the automation development of early sewage treatment. However, with the development of information technology, especially the development of IIoT technology, the traditional monitoring system of sewage treatment plants shows limitations in terms of labor cost and data availability, which cannot give full play to the advantages of multi-terminal sharing and full utilization of data.

In view of the limitations of traditional sewage treatment plant monitoring, this paper takes an actual sewage treatment plant as an example to illustrate how to realize the upgrading of industrial automation with the help of IIoT technology economically. The original automation system of the sewage treatment plant has been effectively upgraded with wide-area monitoring and control functions. This article will describe the transformation and upgrade process in detail.

The structure of this article is as follows. First, the control system model of the sewage treatment plant is analyzed. Then from the perspective of the IIoT, a wide-area wastewater treatment monitoring system based on WeChat Applet and STM32 is proposed. Then, taking a sewage treatment plant as an example, the details of the transformation and the transformation plan are analyzed in detail. Secondly, the hardware part of the monitoring system and WeChat Applet monitoring software are also designed in detail. Finally, the system has been applied to actual automation projects and achieved good results.

## 2. Automation Technology of Sewage Treatment

### 2.1. Process Analysis—CASS Sewage Treatment Process

Traditional sewage treatment processes include the Anaerobic Anoxic Oxic (A2O), the Sequencing Batch Reactor Activated Sludge Process (SBR), and the Cyclic Activated Sludge System (CASS) [6]. The A2O process is simple but has a large circulating water volume. It needs to set in a secondary sedimentation tank and sludge return tank, so it covers a large area and has a large investment. The SBR process uses a time division method to control the aeration tank reaction, and the water purification effect is good, but it has high requirements on the automatic control system, short drainage time, and easy to produce scum and other shortcomings. Therefore, according to the expert demonstration and the owner’s requirements, the CASS process is adopted in the plant. Compared with the traditional processes such as the A2O and SBR, the CASS process has the advantages of simple process flow, low investment cost and being less prone to sludge bulking. The sewage treatment process of the CASS is shown in Figure 1.

### 2.2. Automation Technology

Sewage treatment is a large and complex non-linear system, involving automatic control technology, sensor technology, computer technology and network communication technology. There are more than one thousand devices to be monitored in the sewage treatment plant, and more than 10,000 signals need to be collected. At present, some scholars have studied the remote monitoring system of water quality and ecological environment. Trevathan [7] et al. proposed a smart environmental monitoring and assessment technology (SEMAT). SEMAT is a new remote water environment monitoring scheme, which can scientifically configure the difficulty of water quality environment monitoring and reduce the logistics cost. Capraro [8] et al. take the arid area of Argentina as an example, and have designed a set of remote monitoring and control of irrigation systems by using web technology, combined with special hardware and sensors. Wang, X [9] and others proposed to design an online water treatment monitoring system based on ZigBee and GPRS technology to realize real-time monitoring of data. Xiao, X [10] and others introduced a set of wireless real-time monitoring system, which is used to monitor the running state of underground equipment. The system consists of ZigBee wireless sensor network and real-time remote monitoring unit. Kim, K [11] and others designed a set of remote monitoring system to check whether there are harmful bacteria in the sampling water, and then found qualified water sources. The system adopts Wi-Fi technology, sensor technology, web publishing technology and database technology to realize the online monitoring of water quality. Chen, C [12], et al. proposed a distributed energy management system based on web in order to save energy of buildings and environmental systems. The system uses the PLC as the central controller, and the field equipment uses RS-485 to connect with PLC. By using the industrial software web studio, the host computer software of SCADA is developed, which realizes the data acquisition and monitoring.

The early sewage treatment monitoring systems were composed of PLC and host computer, which supporedt the development of the sewage treatment industry. However, with the arrival of IoT, IIoT and industry 4.0, the early monitoring system technology has been unable to meet the development needs of the monitoring system of the sewage treatment plant, because it cannot achieve remote monitoring and control of sewage treatment plant. At present, there are two kinds of remote monitoring and control technologies to realize the sewage treatment plants, one is to realize online monitoring and control of water quality by using the technology of IIoT; the other is to realize remote monitoring and control of sewage treatment plant by combining the web publishing function of the commercial configuration software and the central controller. Through the web publishing function of the commercial configuration software, the operation of the equipment can only be monitored but not controlled. The web publishing function can only run on the computer, but cannot be monitored by mobile phone, which brings inconvenience to the staff. The monitoring system developed through the IIoT can not only monitor but also control the operation of equipment. As long as staff take out their mobile phones and turn on WeChat, they can monitor the operation of the device in real time.

Up to now, the classic monitoring and control system architecture of the sewage treatment plant generally adopts a hierarchical structure. A typical monitoring and control system architecture is shown in Figure 2. According to the number and characteristics of the monitoring equipment needed by the sewage treatment plant, the monitoring system is divided into three levels: field execution layer, middle layer and application layer. The field execution layer is composed of instruments and equipment, which are responsible for converting field parameters into electrical signals and sending them to a PLC. The middle layer is composed of PLC, which is responsible for processing the signals sent from the field layer and sending them to the host computer after processing. The application layer is composed of monitoring software and server. The monitoring software is responsible for dynamically displaying the on-site running status to the monitoring host, and the server is responsible for saving the running data. The classic monitoring system of sewage treatment can only be monitored locally, and the staff cannot monitor the operation of the equipment in the plant remotely in other places. In recent years, with the rise of IIoT, sensors, computers and other technologies, remote monitoring and control technology has become possible.

### 2.3. IIoT Technology

The digital economy and IIoT are the research hotspots in recent years, bringing great changes and challenges to society. The IIoT organically links sensors, monitoring software and controllers so as to greatly improve industrial production efficiency and save production costs. The IIoT has the following characteristics: the interconnection of equipment, the transmission of equipment data to a cloud platform through Ethernet, and the use of cloud platform data to connect with more equipment. At present, the IIoT is gradually applied in water supply systems, power supply systems, heating and other aspects, but rarely used in sewage treatment plants [13]. With the rapid development of the IIoT, sewage treatment plants need the renovation of the IIoT to make the monitoring system more intelligent, but it cannot work without the support of key technologies. The key technologies of IIoT include the following aspects, shown in Figure 3.

(1)Sensor technology. Sensors are devices for collecting information in real systems, converting analog signals into digital signals and sending them to CPU for processing [14]. Sensor technology includes infrared sensors, gas sensors and Radio Frequency Identification (RFID) technology. RFID technology is a comprehensive technology which integrates wireless radio technology and embedded technology. It has broad application prospects in automatic identification and logistics management. The development of IIoT technology requires more accurate and efficient sensors to achieve rapid detection.(2)Communication technology. The IIoT has many types of equipment and complex working environments. A single technology is often unable to meet communication needs, so it needs to work together with a variety of communication technologies. For example, Wi-Fi, 4G/5G, Ethernet and other technologies of traditional Internet; Bluetooth, ZigBee and other technologies of near distance transmission; RS485, RS232, CAN data line and other technologies of wired transmission; they exist in a group of systems and work together.(3)Embedded system technology. It integrates computer hardware, software programming, integrated circuit, electronic application and other technologies. After rapid development in recent years, WeChat Applets, WeChat subscription numbers, smart phone Apps and other products have emerged.(4)Network and information security technology. Information security is an important guarantee for the normal operation of the IIoT. In the event of a security accident, the system may be paralyzed, or there may be casualties, which may cause significant economic losses. The common security problems of the IIoT include data theft and network intrusion, which are embodied in eavesdropping, password theft, operating system vulnerability attack and phishing.(5)Actuators. In the control system of the IIoT, the actuators are in the equipment layer, responsible for the control of field equipment. Actuators are divided into pneumatic, electric and hydraulic actuators, which are mainly used in sewage treatment industry. A large number of motors and valves, such as lift pump, sludge pump, dosing pump, butterfly valve and check valve, are needed in sewage treatment plant. The rapid development of actuator technology provides support for the development of IIoT [15].

At present, some research work has been carried out on the IIoT. Emiliano sisini [16], et al. introduced Internet of Things (IoT), IIoT and industry 4.0, described the differences between IIoT and traditional industrial control. They shared their latest research results in IIoT, and proposed that IIoT will bring us great opportunities and challenges; Wang, H [17] et al. analyzed the changes brought by the application of IIoT based on the maritime industry in Northwest Norway, and proposed a new framework based on BDA and IIoT technology; Wan Jiafu [18] et al. proposed the software architecture of IIoT based on industry 4.0, discussed the physical layer of network, industrial cloud and intelligent control terminal. Finally, conducted experiments with an intelligent manufacturing terminal analysis; Wang H [19] et al. proposed a data security fusion method of IIoT to ensure the security of IIoT data, through mutual supervision between nodes, and finally proved their security and reliability with game theory. From the above analysis, it can be seen that the IIoT is still in the stage of gradual development and maturity. The current research is mostly based on theoretical exploration, which describes the development, significance, technology and future prospects of the IIoT, and less involves the actual engineering application of the IIoT.

Compared with the traditional sewage treatment monitoring system, the biggest advantage of using the IIoT is to realize the remote monitoring of the sewage treatment plant. The traditional sewage treatment plant uses the PLC controller; PLC controller and module generally need thousands of dollars, but the STM32 control and module of IIoT only tens of dollars. Traditional business software costs tens of thousands of dollars, while WeChat Apps cost hundreds of dollars. Traditional local servers cost tens of thousands of dollars, while cloud servers cost thousands of dollars. Using cloud servers has the advantages of being maintenance free, and of having high security and high reliability. Therefore, using the IIoT can reduce the engineering cost. In addition, Siemens PLC uses PROFIBUS protocol, Schneider PLC uses Modbus protocol, and different PLC uses different protocols. Due to different protocols, it is difficult to realize the IIoT. However, STM and related modules, using serial port protocol, can be compatible with mobile phones and tablets, and easy to realize the IIoT.

In this paper, an actual sewage treatment plant is taken as an example, aiming at the current situation of the automation system of the sewage treatment plant, the wide-area monitoring and control system of the equipment of the sewage treatment plant is renovated by using the IIoT according to local conditions, so that it has a higher cost performance. Through the technical renovation of IIoT, the traditional sewage treatment plant automation system has the characteristics of being safer, more reliable and operating more efficiently.

## 3. Project Overview

A sewage treatment plant in Guangdong Province of China is a renovation project, mainly responsible for the treatment of industrial wastewater and domestic sewage in the center of a city. The current sewage treatment scale is 20,000 tons/day, and the scale after renovation is 50,000 tons/day. In the original project, there is one lifting pump room, one coarse grid and one fine grid, one swirling sand settling pool, two CASS pools and one ultraviolet disinfection pool. After the renovation, there are two lifting pump rooms, two coarse grids, two fine grids, two swirling sand settling pools, four CASS pools and two ultraviolet disinfection pools. The original digital and analog interface equipment shall be reserved, and equipment such as coarse grid, fine grid, sludge pump and various detection instruments shall be added. The original monitoring system adopts the Siemens PLC S7-300 as the central controller, Siemens configuration software WINcc as the host computer, STM32 series single-chip microcomputer as the central controller after renovation, and WeChat Applet secondary development monitoring and control system as the host computer.

In order to achieve efficient remote monitoring and control of the sewage treatment plant monitoring system, the project uses IIoT and other technologies to build a set of sewage treatment remote monitoring and control system by using WeChat Applets and the STM32. Compared with the original project, the improved project has the following advantages:(1)The price of Siemens PLC and host computer WinCC are more expensive, resulting in high project cost, while the price of STM32 single chip microcomputer and WeChat Applet is low, which can save project cost;(2)Siemens host computer WinCC can only be installed in the computer, unable to realize mobile monitoring. The WeChat Applet can run on the mobile phone. As long as the staff scan the QR code and log in, they can monitor and control the running status of the device anytime and anywhere.

The system uses the technology of IIoT, the lower computer uses STM32 single chip microcomputer, the host computer uses WeChat Applet, combined with Tencent cloud server, database, industrial Ethernet and other technologies, to realize the remote monitoring and control of the equipment. The system is divided into monitoring layer, middle layer and field executive layer. The monitoring layer is composed of printers, projectors, local servers, cloud servers, mobile phones and tablet computers; the middle layer is composed of STM32 as the core control cabinet; the field executive layer is composed of sensors and control equipment. In order to enhance the stability of the system, the dual redundant mode of local server and cloud server is selected to prevent data loss caused by local server damage. At the same time, in order to increase network reliability, not only the dual ring network mode is adopted for double-loop network mode, but also the dual redundancy Wi-Fi module block is added for each control cabinet. When the optical fiber is damaged, the system immediately switches to Wi-Fi wireless network; when one of the Wi-Fi modules fail, the system immediately switches to another Wi-Fi module to achieve the effect of multiple redundancies. At the same time, in order to prevent the system from stopping operation due to power outages of the plant, a set of uninterruptible power supply (UPS) devices is added to each control sub-station. The data flow diagram of the monitoring and control system is shown in Figure 4, and the overall architecture of the system is shown in Figure 5.

The system is divided into three control sub-stations due to the large number of monitoring equipment in the sewage treatment plant. Number 1 control sub-station is responsible for the monitoring of pretreatment pool, disinfection outlet pool, sludge storage pool and other equipment, mainly including coarse grid, fine grid, liquid level meter, electromagnetic flow meter; Number 2 control sub-station is responsible for the monitoring of equipment in CASS pool and blower room, mainly including sludge pump, dosing pump, water inlet valve, blower, air flow meter; Number 3 control sub-station is responsible for the monitoring of equipment in sludge dewatering system. The main equipment includes desliming machine, frequency converter, ammeter and voltmeter. The control sub-station collects and processes the field equipment and sensor signals and sends them to the server. The mobile phone is connected with the cloud server to realize remote monitoring and control of the device.

## 4. Field Hardware System Design

The hardware system architecture is shown in Figure 6.

The hardware of sewage treatment monitoring system based on IIoT is mainly composed of STM32 central controller, relay, communication module, clock module, power module and analog -to-digital conversion module. In order to prevent the Wi-Fi module from being damaged and causing the device to be unable to communicate, add a Wi-Fi module in each hardware unit.

### 4.1. STM32 Minimum System Design

In the design of IIoT control system, the controller is one of the most critical parts. Its selection is related to the stable and reliable operation of the system. At present, in the field of IIoT, commonly used controllers include DSP, ARM, STM32 and so on. DSP controller has strong communication function and fast processing speed, but it has high development cost and long development cycle. The ARM controller has large memory and many I/0 interfaces, but it is difficult to develop and has poor flexibility [20,21]. 

The STM32 series single chip microcomputer is a high performance, low cost and low power controller developed by ST Company. The renovation demands of sewage treatment plant compared with the STM32 function are shown in Table 1. Since the STM32 meets the requirements of sewage treatment plant transformation with high cost performance. And the designer is familiar with its development, so STM32 is chosen as the main controller.

The STM32 minimum system is shown in Figure 7. PA9/PA10 and TX/RX form a serial port circuit. PC14/PC15 are connected with crystal oscillator circuit and PH0/PH1 are connected with another one. The BOOT0 is grounded through R4 to prevent the error of reading the internal FLASH code when STM32 starts. PDR_ON is connected with VCC high level through R5. When PDR_ON is high power, it acts as internal power monitor [22,23].

### 4.2. Design of A/D Conversion Module

Analog-to-digital conversion refers to the conversion of analog signals into digital signals for processing. The STM32F4 series is equipped with 12 channels AD conversion, which can handle a small part of analog quantity. However, considering the large amount of control equipment in the system, it is far from enough to use the analog-to-digital conversion of STM32, so 74HC595 chip is used to design the independent analog-to-digital conversion module. The advantage of using an external independent digital-to-analog conversion module in the system is that when replacing the STM32 controller, the external independent digital-to-analog conversion module does not need to be replaced, which improves the efficiency of the design. The 74HC595 chip is used in the circuit design of digital-to-analog conversion module, as shown in Figure 8. SER is the serial data input pin; OE is the enable input terminal, connected to low voltage Q1–Q7 is the data input terminal, Q8 is the data output terminal, and two 74HC595s are connected in cascade.

### 4.3. Digital Quantity Module Design

In the sewage treatment monitoring system, a STM32 controller needs to deal with a large number of digital signals, such as the start, stop, alarm and fault of lift pump, the fault and alarm of instrument. In the practical engineering application, in order to improve the reliability of the system, a STM32 is not directly connected to equipment and instruments, but to optocoupler relay first, and then to instruments and equipment. The most important function of optocoupler relay is isolation. Its function is to prevent strong current from entering weak current in series, so as to protect STM32 from damage [24]. When starting the device, the WeChat Applet sends “1” to a STM32, which receives the signal and sends it to the device to be started after processing. The digital quantity output module is shown in Figure 9. Port OUT0-3 is connected to the output port of STM32. After the signal is isolated by optocoupler, it is output from port Eout0-3. When the digital module is working, a STM32 controller sends a low-level signal to OUT1 port, diode D1 is on, light is sent to triode Q1, EOut0 outputs high-level signal, thus realizing signal isolation [25].

### 4.4. Circuit Design of Wi-Fi Module

After collecting sensor data by the STM32 controller, the sewage treatment monitoring and control system is connected to the cloud server through the Wi-Fi module. The WeChat Applet exchanges data with the cloud server through Ethernet, so as to realize the remote monitoring and control of sewage plant equipment. The Wi-Fi module selects the ESP8266 model, which has superior performance, low price and strong universality. It not only integrates analog transmission/reception, but also integrates I2C and other communication interfaces, providing good technical support for industrial interconnection IoT scheme [26]. ESP8266 Wi-Fi has three modes: Soft AP, Station and Soft AP + Station [27]. It adopts the Soft AP wireless access point mode with the STM32 controller and cloud server. When the Wi-Fi adopts the wireless access point mode, it transmits a hot spot to realize remote communication between Wi-Fi module and cloud server. The circuit connection of Wi-Fi module is shown in Figure 10.

## 5. Development of Wide-Area Monitoring and Control Software by WeChat Applet

At present, the monitoring software of the sewage treatment plant basically adopts configuration software for secondary development. The common configuration software includes Factory Talk of Rockwell Company in the United States, WINCC of Siemens Company in Germany, King View of Ya Kong Company in China and Force Control of Li Kong Company [28]. This configuration software supports the local data monitoring of the sewage treatment plant, but they can only be monitored locally in the plant, not remotely, and cannot adapt to the wide-area data monitoring under the background of IIoT [29]. In recent years, with the development of computer technology and the rise of the IIoT, WeChat Applets and Android Apps are more and more widely used [30]. This project selects WeChat Applet secondary development monitoring and control system, which has the advantages of low development cost, convenient use, and remote real-time online monitoring and control system.

### 5.1. Overall Architecture of WeChat Applet Software System

The software system architecture is designed according to the sewage treatment process, which is divided into the main interface, water inlet system, sand removal system, reaction system, blower room, disinfection system, dehydration system and power system [31]. The main interface of the system displays the process flow and general situation of the sewage treatment plant. The water inlet system is mainly responsible for the equipment monitoring of the coarse grid and the lifting pump room. The sand removal system is mainly responsible for the monitoring of the fine grid and the sand removal equipment. The reaction system is mainly responsible for the monitoring of the four CASS pools equipment. The blower room is mainly responsible for the monitoring of three blowers. The disinfection system is mainly responsible for the monitoring of the ultraviolet ray disinfection pool equipment. The dehydration system is mainly responsible for the monitoring of dehydrators and dosing equipment. The power system mainly includes the collection of voltage, current, temperature, power and other parameters such as the high-voltage distribution room, low-voltage distribution room and transformer. The overall architecture of the wide-area monitoring and control software system is shown in Figure 11.

#### 5.1.1. Design of Equipment Control System

The sewage treatment plant has a large amount of equipment to be controlled, such as the lifting pump, coarse grid, blower and fine grid, which usually need to monitor the start, stop, failure and local/remote of the equipment. When controlling the equipment, the operator first logs in to the monitoring software and selects the corresponding monitoring interface. For example, to control the remote automatic operation of number 1 lifting pump, select the water inlet system monitoring interface, select number 1 lifting pump, click remote automatic operation, and the lifting pump can automatically operate according to the set program.

There are several ways to control the equipment. In the “Field control” mode, the equipment can be controlled by the control button of the field equipment operation box; in the “Remote manual control” mode, the start and stop of the equipment can be controlled on the WeChat Applet monitoring interface; in the “Remote automatic control” mode, after clicking start on the WeChat Applet monitoring interface, the equipment follows the set process the sequence runs automatically. The flow of equipment control system is shown in Figure 12.

#### 5.1.2. Design of Monitoring System

The sewage monitoring module mainly includes a pH value measuring instrument, liquid level meter, electromagnetic flow meter and other instruments. The function of the pH meter is to detect whether the pH value of the water quality is within the specified range; the function of the level meter is to measure the liquid level, so as to decide whether to start the lifting pump; the function of the electromagnetic flow meter is to measure the flow of sewage. When the monitoring system is working, first the system is initialized, and then the STM32 receives commands from WeChat Applet. When the system receives the calibration command, STM32 sends the command to the calibration module, such as the calibration of the reference liquid level of liquid level gauge and the initial value of electromagnetic flow meter. When the system receives the upload command, STM32 sends the command to the upload module, such as the instantaneous value and cumulative value of liquid level gauge. When the system receives the control command, STM32 sends the command to the control module, such as through frequency control of the frequency converter, which further controls the speed of the lifting pump and the flow of the monitoring module, as is shown in Figure 13.

### 5.2. Cloud Platform and Database Design

#### 5.2.1. Cloud Platform Construction

The traditional industrial control field adopts the distributed control system, whose server is built by the local servers, and transmits the data to PLC through the bus or Ethernet for analysis and processing. With the arrival of the era of big data and the rapid development of cloud platform technology, traditional SCADA has difficulty processing the massive data and realizing remote storage, while the cloud platform technology is able to store data remotely and has a powerful data processing function, which is applicable to the IIoT [32]. At present, there are many enterprises providing cloud platform technology on the market, such as Alibaba cloud, Tencent cloud and Baidu cloud. Considering that Tencent cloud and WeChat Applet are products of Tencent, their compatibility will be better. Tencent cloud platform is selected in this scheme. The differences between Alibaba cloud, Tencent cloud and Baidu cloud are shown in Table 2.

The Tencent cloud platform system is divided into three layers: application layer, platform layer and resource layer [33]. The SaaS (Software - as - a - Service) in the application layer mainly faces users and enterprises, and provides them with open interfaces for convenient access to cloud data. The PaaS (Platform - as - a - Service) in the platform layer is the core part of cloud platform, which mainly provides programmers with technology to develop applications. The IaaS (Infrastructure - as - a - Service) in the resource layer mainly integrates resources such as network, storage and operation into a dynamic virtual processing center, which can be fast, with reasonable distribution and recovery when users need them, as shown in Figure 14.

#### 5.2.2. MySQL Database Design

The database system is the core part of monitoring the software data storage, which is responsible for storing various parameters generated by the system operation, such as user login information, equipment operation data, instrument real time measurement value and cumulative value. At present, common databases include MySQL, Oracle and Access. The MySQL database is adopted in this project, which has the characteristics of less occupied resources, good security performance and opens source code, is suitable for industrial Internet control system [34]. According to the functional requirements of the control system of sewage treatment pump station, the design database mainly consists of user management, alarm list, historical trend, real-time trend, report management and other data tables. The user management is mainly responsible for the storage of user information, and the alarm mainly records the alarm records caused by various events. The overall design of the database is shown in Figure 15.

### 5.3. Implementation of WeChat Applet Monitoring System

The host computer of sewage treatment monitoring system is developed by WeChat Applet to realize the functions of equipment monitoring, fault alarm, historical data, real-time data and report query. In the process of developing the WeChat Applet monitoring interface, it mainly includes three aspects: the establishment of the WeChat Applet development environment, the design of the monitoring interface and the data collection.

#### 5.3.1. Construction of WeChat Applet Development Environment

To develop the WeChat Applet, we need to first register an account on WeChat public platform, obtain a unique project management Applet ID, and then download the development tools on the official website. WeChat Applet development mainly includes code editing, code debugging and Applet simulator. The directory structure developed includes js, json, wxml and wxss files, corresponding to programming logic, parameter configuration, page structure and code style. The interface of WeChat Applet import project is shown in Figure 16, and the interface of WeChat Applet development is shown in Figure 17.

#### 5.3.2. Design of WeChat Applet Monitoring Interface

The WeChat Applet monitoring interface includes system main interface, fault record, historical data and report management. The monitoring interface adopts WeUI framework, which is the basic style library consistent with WeChat vision, and adopts HTML5 and CSS3 technology, which has the advantages of simple development and quick system response. The code of the development monitoring interface is as follows:

<view class=“container loglist”><! -l Use Wx: for loop to list real-time data-->

<block wx: for = “{{ current Arr}} ”

wx: foritem = “log”

wx: key=“*this”><!—current Arr array-->

<view class=“WeUI-cell”><!-index Array index, location, state Representative monitoring location and status->

< /block >

< /view >

The development of WeChat Applet monitoring interface is shown in Figure 18.

#### 5.3.3. Design of Data Acquisition System

The data acquisition system needs to establish a database to store the corresponding data, such as the liquid level value of the liquid level meter, frequency of the frequency converter, cumulative flow and instantaneous flow of the electromagnetic flow meter. After all kinds of data are collected, they are saved in the server. At the same time, daily report, monthly report and annual report are established for operators and managers to query. This design uses MySQL database to establish three tables: the T-User table to store user information; the T-Dev table to store online monitoring data; the T-Pow table to store fault information. The http Request is used for the interaction between WeChat Applet monitoring interface and background data. WeChat Applet actively sends request command to background server to realize data communication. The differences between T-User, T-Dev and T-Pow are shown in Table 3.

#### 5.3.4. Design of Security Certification System

With the rapid development of Internet technology, more and more devices have access to the Internet, and security issues are increasingly prominent. Security is the key to the IIoT, which is the premise to ensure the stability and reliability of the system. If the system is intruded by hackers, the system will stop running, or possibly cause personal injury and heavy economic loss. At present, common security problems include internal security and external security. The internal security refers to computer virus infection, illegal person login and system failure. The external security refers to remote invasion of hackers, attacks on local servers or cloud servers and phishing [35,36]. The security certification system is shown in Figure 19.

(1)User authentication

User authentication is one of the simplest and most reliable ways of gaining system security. Only users with correct user name and password can access the system. According to the characteristics of the sewage treatment monitoring system, three login modes are designed for managers, operators and tourists. The manager has the highest authority and can perform all operations on the system, including authorizing other operators, setting passwords, add or remove users; the second is the operator’s authority, which can not only monitor the operation of the equipment, but also control the equipment; the visitor’s authority is assigned to the visitors, who can only view the operation of the equipment, not control the equipment.

(2)MAC Address Binding

The Mac is short for physical address. Each device has its own physical address. There are a large number of equipment connected to the network in the sewage treatment plant, and the system identifies whether the equipment is normal or not through the physical address. When designing a system, the physical address of each device is bound to the system. Only devices bound to the system can be recognized by the system. By binding the physical address, the security of the system is greatly enhanced.

(3)System Log

The System log is to record the information of hardware and software operation, such as system startup and exit records, user operation records and alarm records. The administrator can view the system log to view the operation status of the system. The log of sewage treatment control system includes work log, application log and security log. The work log refers to the login and logout of the staff and the operation records of different operators; the application log refers to the operation status of the program, the start and stop status of the equipment; the security log refers to the alarm records and illegal access records of the equipment.

## 6. System Test and Discussion

Among all the performance of the monitoring system, stability and reliability are the most important. If the system fails and stops running, it will cause great losses. Therefore, the system test in the factory is an essential link. Before the test, first set up the test platform, connect the STM32 central controller, cloud server, Ethernet, signal generator, and select different models of mobile phones for the test. The test items include functional test, communication test and reliability test. 

### 6.1. Functional Test

During the experiment, open the WeChat scanning QR code and enter the WeChat Applet monitoring main interface. Then click login, enter the user name and password to obtain the operation authority. After that, Click the button of the lifting pump room, set the script program to “1”, enter the monitoring interface of the lifting pump room, and display the six lifting pumps on site. Furthermore, click the number 1 lift pump button to enter the fine grid control interface. Click “Remotely”, the variable G101_AM==1, the fine screen will run automatically remotely; click “Start Up”, the variable G101_RUN==1, the fine screen will start; set the fine screen fault variable G101_FAULT==0, the WeChat Applet will display “Malfunction”. The interface of system test part is shown in Figure 20.

According to the above test results, the system adopts the technology of the IIoT to transform the existing field automation equipment at a low cost. The secure communication is realized by the WeChat Applet, and the wide-area monitoring and control of the sewage treatment plant are realized. Each control button can control the startup and shutdown of the equipment, and the instrument displays such as the frequency converter and the flow meter which can collect the instrument data in real time. The whole monitoring system can run normally, collect data and control equipment in real time.

### 6.2. Fault Test

A series of biochemical reactions are needed in the sewage treatment plant to purify sewage. Biochemical reaction in the CASS tank is one of the links of wastewater treatment, via adding medicine to the Cass tank, and then filling air into the blower to make the medicine and sewage fully react. There are four blowers in the blower room—three working and one on standby. The CASS tank is 6.0 m deep, equipped with a liquid level gauge, and the number of blowers to be started is determined according to the liquid level of CASS tank. When the liquid level of the CASS pool rises and is more than or equal to 2 m, it starts one blower; when the liquid level is more than or equal to 3.5 m, it starts two blowers; when the liquid level is more than or equal to 4 m, it starts three blowers; when the liquid level is more than or equal to 6 m, the system will give an alarm. When the liquid level of the CASS pool drops and the liquid level is less than or equal to 2 m, it closes one blower; when the liquid level is less than or equal to 1 m, it closes two blowers; when the liquid level is less than or equal to 0.5 m, it closes three blowers; when the liquid level is less than or equal to 0.3 m, the system gives an alarm. 

The discharge water quality indexes of sewage treatment plant include potential Hydrogen (pH), Chemical Oxygen Demand (COD), and Suspended Solid (SS). The effluent water quality indexes are greatly affected by the process, especially in the biochemical treatment of CASS pool. The biochemical tank is filled with air through the dosing and blower of the dosing pharmacy to fully react to achieve the purpose of purifying sewage. However, in the actual operation, sometimes the blower stops. For example, the operation time of the blower is too long, and the action of the thermal relay causes the blower to stop running, which will seriously affect the effluent quality index. The sewage treatment plant is usually operated 24 h a day and the staff need to be on duty at night. With this system, the staff can go home to sleep at night. When the device fails, it can remotely control the operation of the device in the normal mobile phone. For example: blower 1 alarms and stops. The mobile phone client of the staff receives the alarm signal, turns off the number 1 blower, and starts the standby blower, so as to make the equipment operate normally and ensure that the effluent water quality reaches the standard. The blower failure test is shown in Figure 21.

The blowers were tested for failure in the laboratory. Open STM32 control program, set number 1 blower failure variable B601_Falut = 1, blower display failure, mobile monitoring client receives alarm signal. Clear the alarm signal in the mobile client, stop the operation of blower 1, and start the standby blower. On the software program, it is monitored that B601_Falut = 0, B601_Stop = 1, B604_Run = 1. The blower is an important piece of equipment for the biochemical reaction. Once it breaks down, it will affect the quality of the sewage. When there is no remote monitoring system, if there is a fault at night, the staff need to spend about 45 min getting to the sewage plant by public transport, affecting the effluent. After using this system, the staff can deal with it in 3 min, ensuring the quality of the effluent. The COD instrument range is (0–100) mg/L, normal water quality standard is (40–60) mg/L; SS instrument range is (0–100) mg/L, normal water quality standard is (10–20) mg/L, pH instrument range is (0–14) and normal water quality standard is (6–8). In the experiment, when the CASS tank level is 0.8 m, only one blower is running. The influence of the blower on the COD of effluent water quality is shown in Figure 22, the influence of the blower on the SS of effluent water quality is shown in Figure 23, and the influence of the blower on the pH of effluent water quality is shown in Figure 24.

It can be seen from the experimental results that when the blower is broken, the original monitoring system does not have a wide-area monitoring function and cannot handle the faults that occur in real time, and the water quality treatment effect is not good. After adopting this method, the system has remote monitoring and control capabilities. When the blower fails, the fault information will be transmitted to the operator in real time through the communication network, and the operation and maintenance personnel can restore the faulty system as soon as possible, and the water quality treatment effect is hardly affected. Because when the original monitoring system fails, the staff are asleep at home at night, and it takes half an hour to get to the plant by means of transportation. Through the modified system, the staff only need to be at home to monitor the operation of the equipment. The COD is the value of chemical oxygen demand. The shutdown of the blower has the greatest impact on the COD, and the COD value of sewage increases. The SS refers to the solid substance suspended in the sewage. When the blower stops running and no air are injected into the sewage, it is beneficial to the sedimentation of the solid substance. Therefore, the SS value is decreasing. The PH refers to the ratio of the total amount of hydrogen ions in the solution to the total amount of substances. The shutdown of the air blower has little impact on the pH value, and the pH value of the sewage slightly increases.

### 6.3. Discussion

In the test phase, all functions can be basically realized, but there are also some problems, for example: It can be seen from the above tests that after adopting the method proposed in this paper, the sewage treatment plant has realized wide-area monitoring and achieved better control effects. Without too much investment, the wastewater treatment plant realizes performance optimization based on the IIoT. Of course, from the perspective of the IIoT to in-depth development, the design of this paper also faces the following challenges.

(1)Number 3 lift pump cannot be started. The tester clicks number 3 lifting pump on WeChat Applet monitoring software to start, and number 3 lifting pump fails to start. At first, the tester thought that the lifting pump was broken, and replaced another normal number 2 lifting pump with a replacement method. The test results show that number 2 lifting pump cannot be started, which proves that the lifting pump is normal. The tester checked the control line and power line of number 3 lifting pump and found that they were not wrongly connected. Then, the tester opens the STM32 control program and finds the starting variable p103_run of number 3 lift pump. After careful inspection, it is found that p103_run = 0, which proves that STM32 has not received the control command from WeChat Applet. The tester opened the development interface of WeChat Applet, found the start button of number 3 lifting pump, and found that the variable was incorrectly written as “p103_ rum”, resulting in the failure to control the operation of number 3 lifting pump. After changing the variables, the test was carried out again, and number 3 lifting pump was able to operate normally. The main reason for this error is that the designer carelessly wrote “p103_ run” as “p103_ rum”, resulting in the failure to start the number 3 lifting pump.(2)The liquid level meter value displayed on the WeChat Applet monitoring software is not correct. The liquid level gauge is used to measure the depth of the pool by the principle of ultrasonic reflection. The monitoring software cannot display the correct level value, which proves that the monitoring software and the level gauge can communicate normally. The tester first checks the level gauge and its wiring, and finds that the level gauge can work normally, and the display screen of the level gauge also has correct data display. Then, the tester opened the STM32 monitoring program, and found that the LD101 variable of liquid level meter value had data display. It can be seen from the test that the liquid level gauge can work normally and communicate with STM32 normally. Opening the WeChat Applet, the tester found that the instrument signal (0–20 mA) was converted to (0–65535), and there was an error in the conversion range. The liquid level value is an unsigned integer, and the correct conversion value should be (0–32767). After changing the conversion value, the level gauge can display correctly. The reason for this error is that the instrument range conversion value input is wrong.

At present, one iPhone and two Android Phones were used in the laboratory test. However, in the actual operation process, dozens of staff may log in the WeChat Applet monitoring and control system at the same time, which is a severe test for the system.

## 7. Conclusions and Future Works

In view of the problem that the monitoring system of the sewage treatment plant cannot be monitored remotely at present, this paper studies a set of sewage treatment monitoring and control system based on the technology of IIoT. The system uses sensors to collect field operation parameters, STM32 single-chip microcomputer control equipment, Tencent cloud server to store data and WeChat Applet to develop the monitoring and control software, which realizes wide-area monitoring and control of equipment in the factory. The result of the laboratory test shows that the system can monitor and control all kinds of equipment and instruments of sewage treatment on line, and that the operation is stably and reliable, achieving the expected effect. The main innovation of this paper is to develop a wide-area monitoring and control system by using the WeChat program, which can provide reference for technicians to develop the remote monitoring and control system of sewage treatment. The summary of this paper is as follows:(1)An STM32 controller is used in this project, which is responsible for collecting sensor data and controlling execution equipment. The traditional sewage treatment monitoring system uses a PLC controller, but a PLC controller is expensive. A set of PLC controllers is about several thousand dollars, while a set of STM32 and supporting modules are only about ten dollars. Therefore, replacing PLCs with STM32s can reduce engineering costs.(2)The host computer adopts the secondary development of the WeChat Applet to realize the remote monitoring of the equipment. Traditional commercial software usually needs several gigabytes of hard disk installation space, and other supporting software needs to be installed before installation. The installation process is complex and error prone. The WeChat Applet monitoring system of this project is based on WeChat big data platform. Users only need to take out their mobile phones, open WeChat and scan QR code to monitor the operation of equipment in the factory.(3)Remote monitoring of the sewage treatment system is realized. Data is transferred to the cloud platform through the flexible IIoT. It can remotely monitor the operation status of the sewage treatment system in real time, and perform online fault repair, which greatly facilitates the work of operation and maintenance personnel and improves the operation efficiency of the sewage treatment plant.

At present, the project is only a small sewage treatment plant, which is transformed into the IIoT, realizing the remote monitoring of the equipment in the plant. With the development of the computer hardware, software and sensor technology, the technology of IIoT will be more mature and the system will be more stable. In the near future, through the IIoT, the interconnection of all sewage treatment plant data will be realized. Through the IIoT, the government departments can monitor the water quality discharge of the sewage treatment plant in real time and online, so as to prevent the occurrence of illegal discharge of sewage.

## Figures and Tables

**Figure 1 sensors-20-02175-f001:**
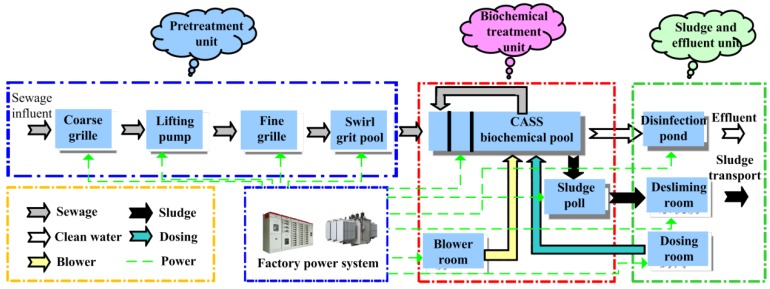
Cyclic Activated Sludge System (CASS) process sewage treatment flow chart.

**Figure 2 sensors-20-02175-f002:**
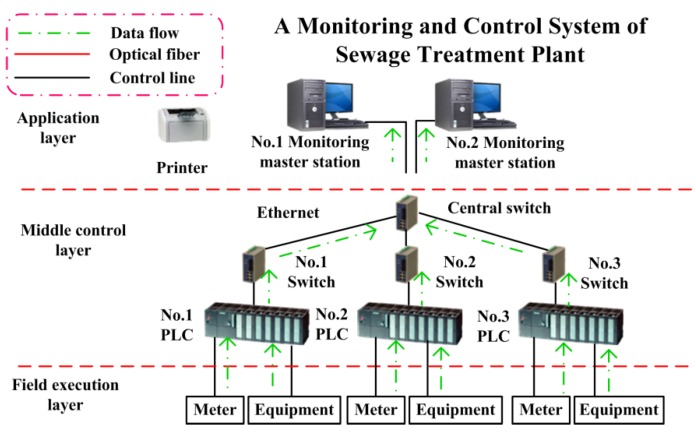
Classic sewage treatment system architecture.

**Figure 3 sensors-20-02175-f003:**
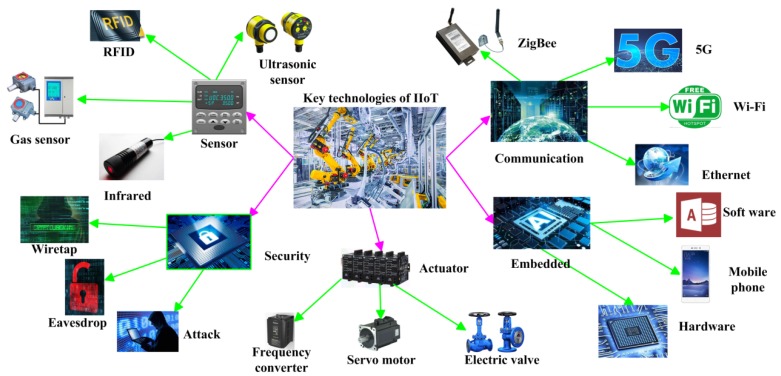
Key technologies of the Industrial Internet of Things (IIoT).

**Figure 4 sensors-20-02175-f004:**
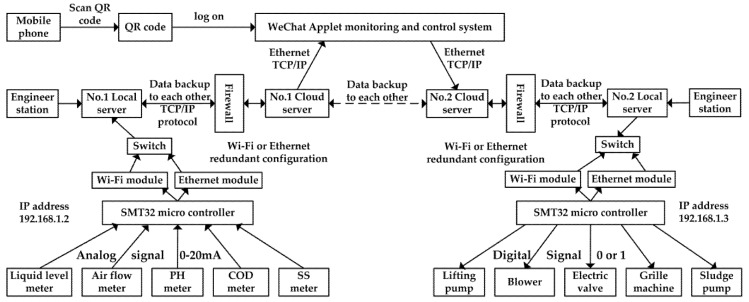
Data flow diagram of monitoring and control system.

**Figure 5 sensors-20-02175-f005:**
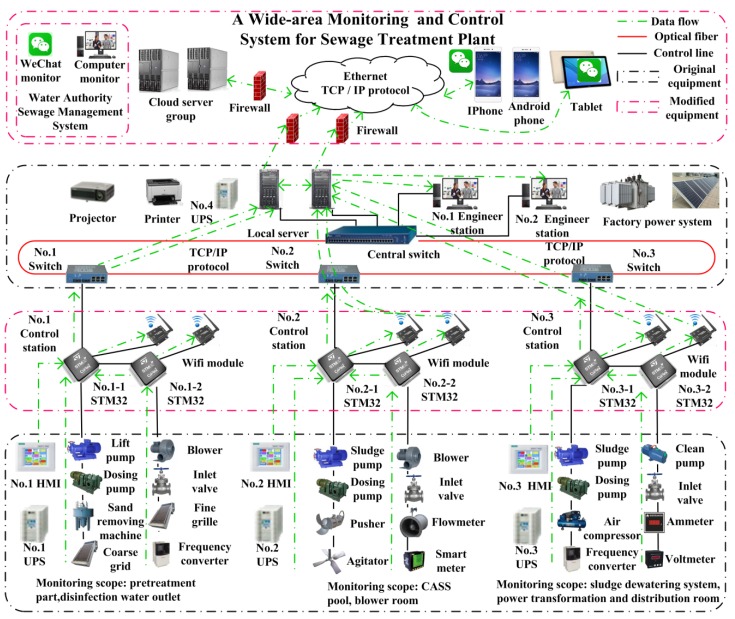
Overall system architecture.

**Figure 6 sensors-20-02175-f006:**
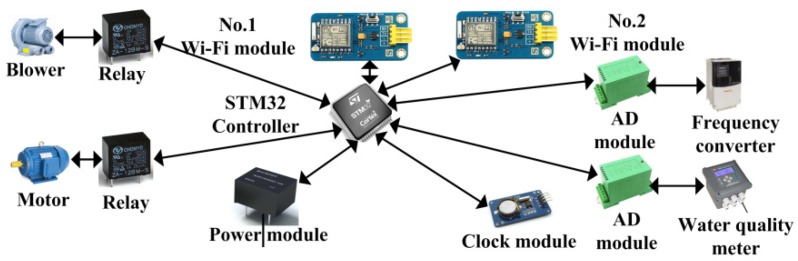
Hardware system architecture.

**Figure 7 sensors-20-02175-f007:**
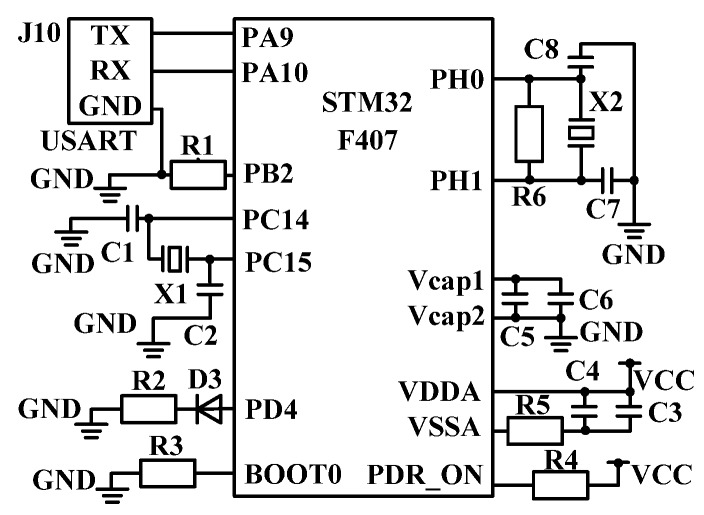
Minimum system diagram of STM32.

**Figure 8 sensors-20-02175-f008:**
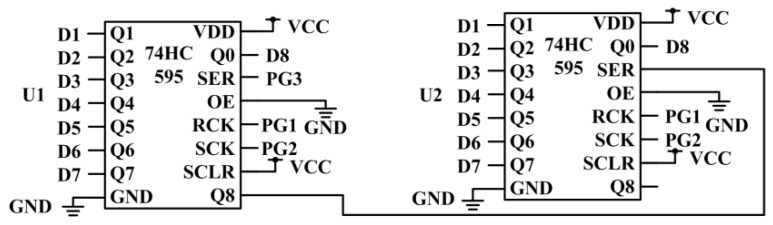
Circuit diagram of A/D conversion module.

**Figure 9 sensors-20-02175-f009:**
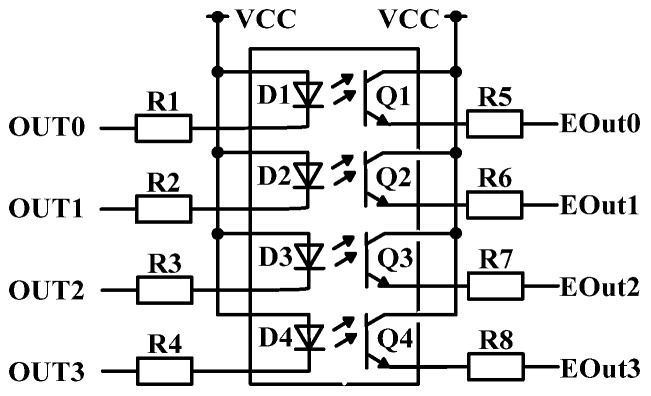
Design of digital quantity output module.

**Figure 10 sensors-20-02175-f010:**
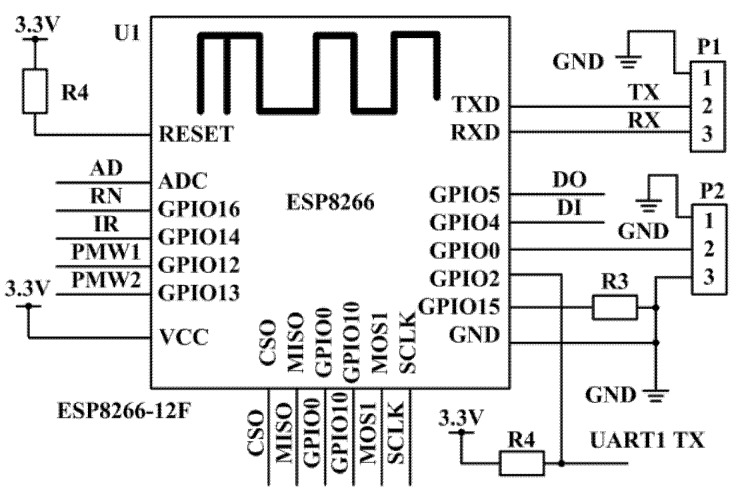
Circuit connection diagram of ESP8266 Wi-Fi module.

**Figure 11 sensors-20-02175-f011:**
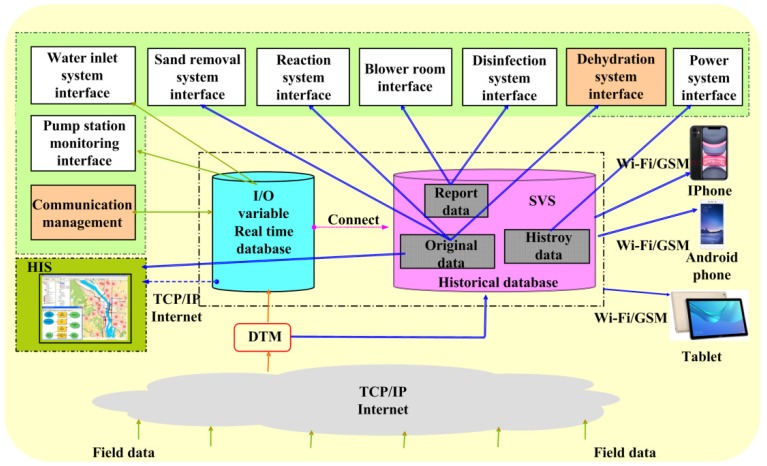
Overall architecture of wide-area monitoring and control software system.

**Figure 12 sensors-20-02175-f012:**
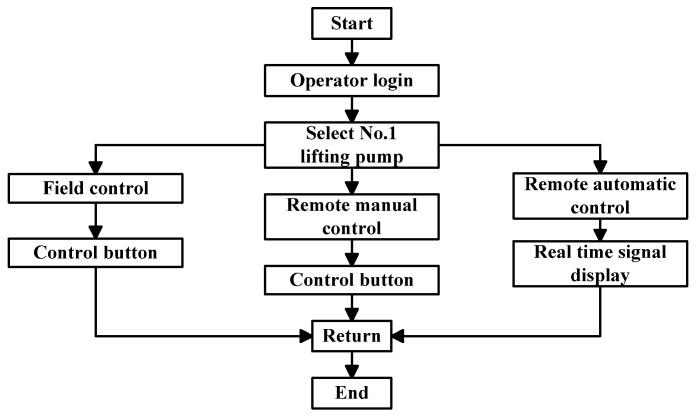
Flow chart of equipment control system.

**Figure 13 sensors-20-02175-f013:**
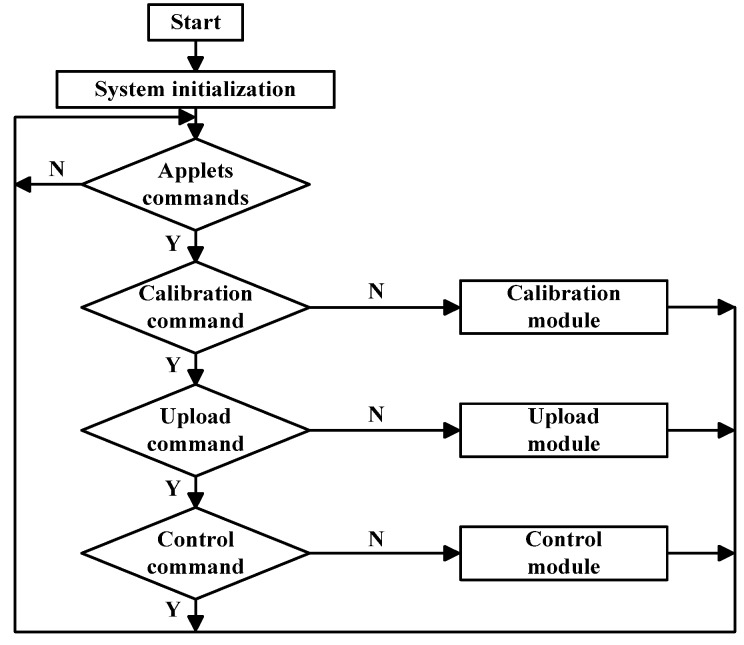
Flow chart of monitoring module.

**Figure 14 sensors-20-02175-f014:**
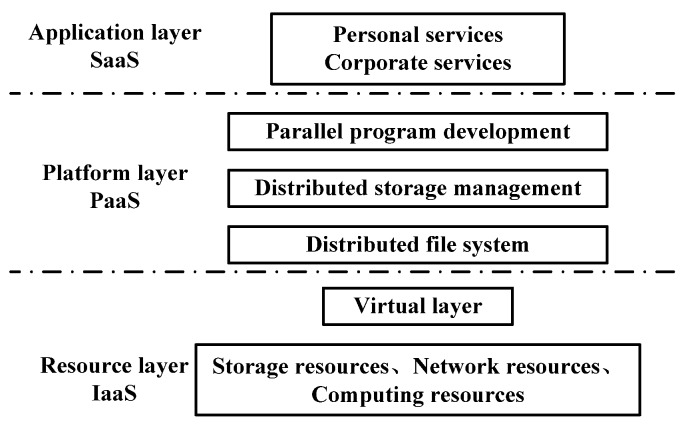
Cloud platform architecture.

**Figure 15 sensors-20-02175-f015:**
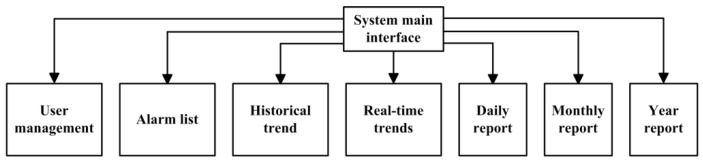
Overall design of database.

**Figure 16 sensors-20-02175-f016:**
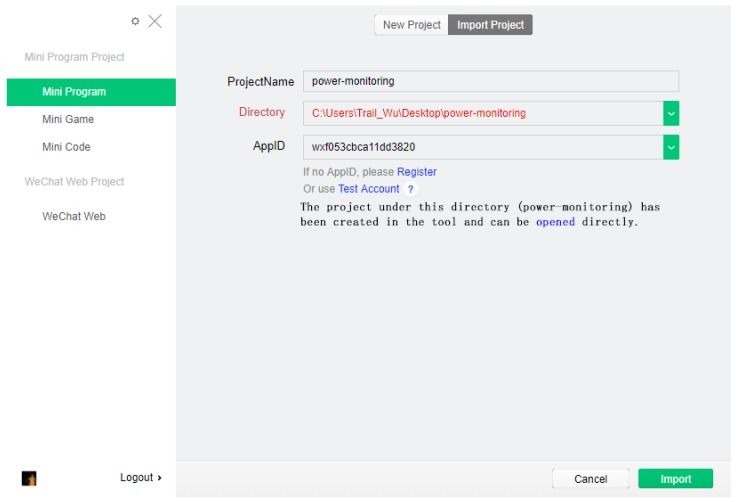
WeChat Applet import project interface.

**Figure 17 sensors-20-02175-f017:**
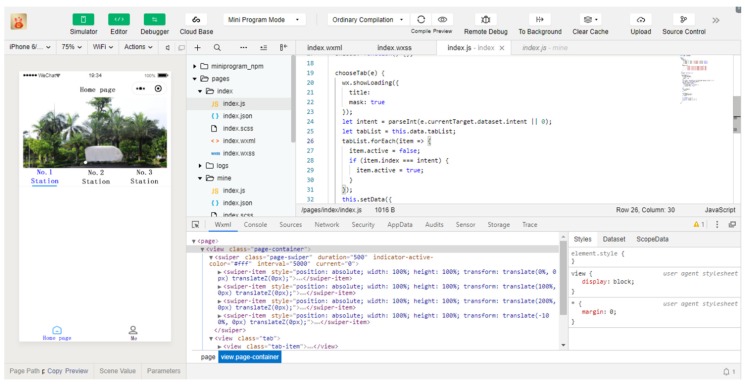
WeChat Applet development interface.

**Figure 18 sensors-20-02175-f018:**
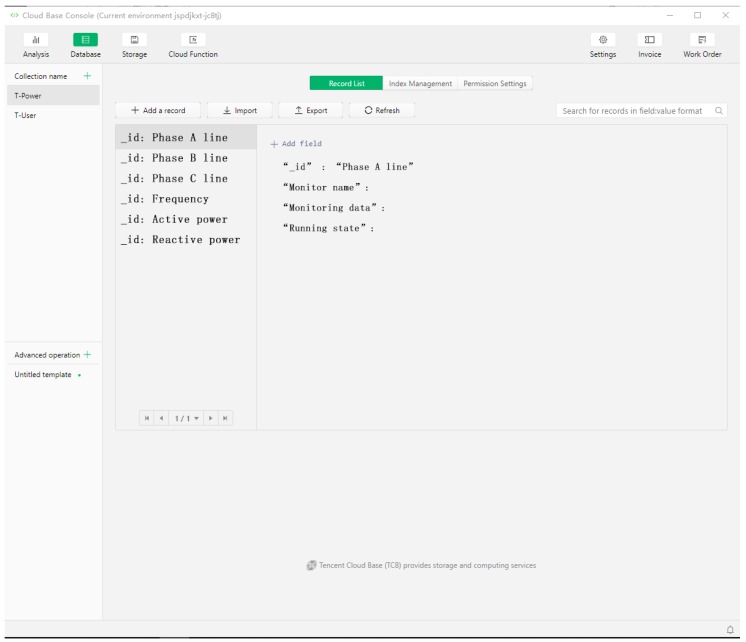
WeChat Applet monitoring interface.

**Figure 19 sensors-20-02175-f019:**
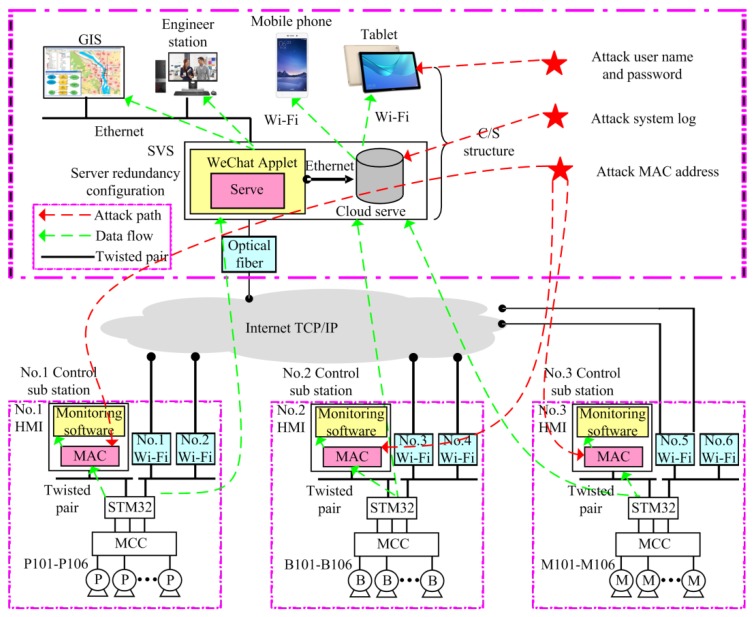
Security certification system.

**Figure 20 sensors-20-02175-f020:**
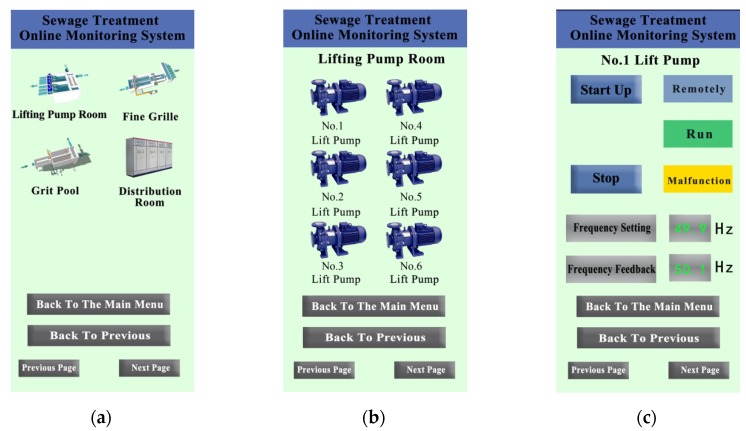
WeChat Applet development interface: (**a**) sewage treatment online monitoring system main interface, (**b**) lift pump room interface, (**c**) number 1 lift pump interface.

**Figure 21 sensors-20-02175-f021:**
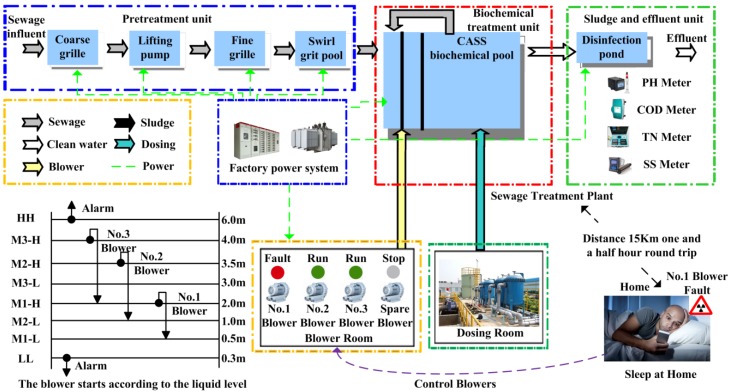
Schematic diagram of blower failure test.

**Figure 22 sensors-20-02175-f022:**
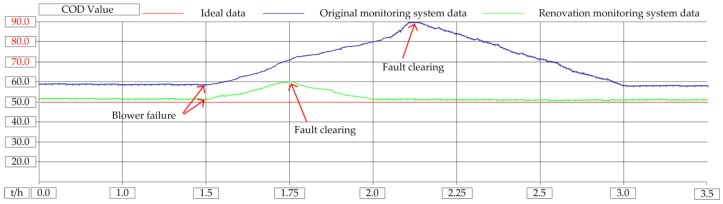
Influence of blower failure on COD of water quality.

**Figure 23 sensors-20-02175-f023:**
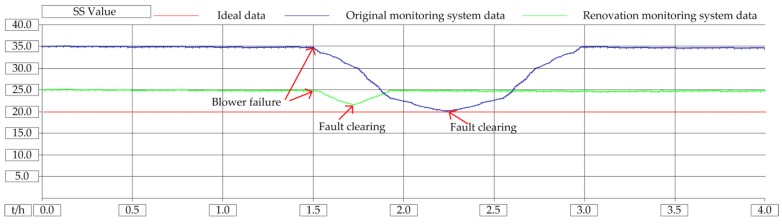
Influence of blower failure on SS of water quality.

**Figure 24 sensors-20-02175-f024:**
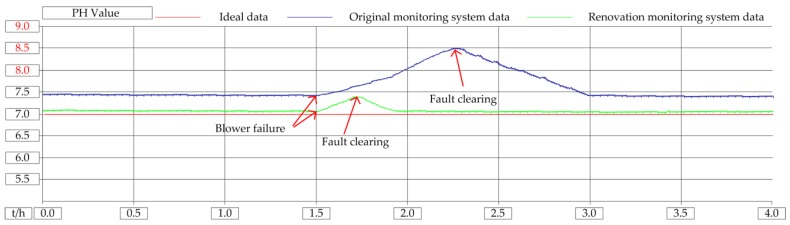
Influence of blower failure on pH of water quality.

**Table 1 sensors-20-02175-t001:** Sewage treatment plant renovation demand and STM32 functions comparison table.

Serial Number	Renovation Demand of Sewage Treatment Plant	STM32 Function	Meet the Requirements?
1	The controller is required to have the function of fast data processing	ARM 32-bit Cortex-M4 core is adopted, and the maximum operating frequency is 168 MHz	Yes
2	It is required to have analog input/output channel and can be expanded	It has 16 AD ports and can be connected with AD module	Yes
3	It is required to have digital input/output channel and can be expanded	It has hundreds of I/O ports and can be expanded	Yes
4	It is required to have memory for storing programs and be able to expand	1 MB flash memory, 192 KB SRAM, and memory expansion	Yes
5	It is required to have network communication function	It can be connected with Wi-Fi module and Ethernet module	Yes

**Table 2 sensors-20-02175-t002:** Alibaba cloud, Tencent cloud and Baidu cloud comparison table.

Name	Market Share	Development Company	Common Fields
Alibaba cloud	High	Alibaba company	Taobao/Tmall and other e-commerce platforms
Tencent cloud	Medium	Tencent company	WeChat service account; WeChat Subscription account; WeChat Applet
Baidu cloud	Low	Baidu company	Website, network storage, artificial intelligence, etc

**Table 3 sensors-20-02175-t003:** T-User, T-Dev and T-Pow comparison table.

	T-User	T-Dev	T-Pow
Name	User information	Operational data	operation state
Data type	Char	Int	Int
Function	Store user information	Store online monitoring data	Store fault information

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
