# Peer review of "Renovation of Automation System Based on Industrial Internet of Things: A Case Study of a Sewage Treatment Plant"

_sensors, 2020, doi:10.3390/s20082175_

Round 1

Reviewer 1 Report

In this paper, the authors have presented a very important application that really deserves special treatment in a sewage treatment plant. The idea of the application is important, especially since it fits in the context of the digitization of factories known today by industry 4.0 /5.0 or society 5.0.

The paper is well written, but unfortunately it is not detailed and lacks too many details that will highlight all the work.

I invite the authors to follow these recommendations:

  • The introduction is very short, it lacks details (more details in the second section which should be in the introduction).
  • The automation of the wastewater treatment service is well detailed in the second section, with a critical analysis. It lacks on this part the technologies used with a comparison to justify the choice.
  • The idea of the project is well detailed: equipment used, interconnection (Figure 4 is well detailed despite the fact that the system is on 4 layers and not 3).
  • The authors present in the fourth part the architecture of the hardware system. In this part, it is clear that the authors used the ESP8266/32 platform in order to send information via WiFi to the next layer. I do not see the usefulness of all the hardware discription on STM32 and ESP8266 especially as these platforms are available on the market and they are not expensive. something else, wifi communication deserves a PUB/SUB communication protocol adapted to the information sent, of which the authors did not specify anything. There are also some missing figures of implementation of all this and especially the adaptation of the information coming from the S7 automaton.
  • Section 5 presents the development of the supervision and control platform based on Wechat Applet. I didn't really find the implementation except some flowcharts and a Wechat Applet development figure. My question: is the application developed or still development? They are or the results? The authors have thought to add the security part which I find important
  • The test of the system is really too modest. than the GUI without any results. I invite the authors to validate and add the results. Above all, we must highlight all the work even more through the results.

I invite the authors to review the structure of the paper, to emphasize even more the idea of the paper and especially to add the figures of the implementations and the results.

Reviewer 2 Report

The authors present the design and implementation of an automation system for a sewage treatment plant under the scope of the IIoT. They clearly describe the motivation for the system and the projected improvements and advantages compared to the current implementation.

The first two sections of the document have no major issues. English grammar should be reviewed to correct some structures and more references could be included, especially when referring to the IoT and IIoT.

Section 3 sets the context for the system, describing what is being improved and how it is done. However, the system architecture presented in figure 4 includes each of the elements in the system and is at some point complicated to understand. I recommend to first include a high-level architecture (in the form of block diagrams) and then advance to this level of detail for each of the blocks.

My main concern is with section 4. It seems more like a technical report with information from datasheets and it is not totally clear if some of the diagrams are original from the authors or from the manufacturers of each device.

Similar concerns apply to section 5, especially with the criteria for platform selection; they are mentioned but a comparison, perhaps in a table, would make for a better understanding. The design of the database shown in figure 14 is more like what is going to be obtained or created from the data stored in the database, but a table model showing relationships would make a better fit for this design, at the level that intelectual property agreements allow. There is a little more detail of the tables in section 5.3.3 that can be included in this diagram.

The test in section 6 is presented in the form of a laboratory report, but it lacks a description of the test plan and objectives, and, more important, the description and discussion of the results. i.e. How many times was the experiment repeated?, Did it return correct readings every time?, Was there any difference when using different devices?, How many devices were used at once?, among others.

There is a sentence at the end of this section that seems a bit strange, when authors state that "thousands or even tens of thousands of staff may log in the WeChat Applet monitoring and control system at the same time". Is this accurate? I don't know the context of the plant or the city where it is located but it seems like a very high number of concurrent users for a control system.

I also think that conclusions should be improved.

Reviewer 3 Report

The paper presents actual real-life results of renovation an automation system - a sewage plant - through the industrial IoT approach. This is the main contribution of the paper. The elements are not now, but the fact that this paper reports on an actual case study of renovation with an IIoT approach is very motivating.

The main weakness of the paper is its language. English should be improved. Examples (just a few; the text should be carefully corrected from head to toe):

  • "the whole monitoring system is composed of local data by upper computer and PLC." - this is very vague
  • "include Anaerobic Anoxic Oxic (A2O), Sequencing Batch Reactor Activated Sludge Process (SBR), Cyclic Activated Sludge System (CASS)" -> include the Anaerobic Anoxic Oxic (A2O), the Sequencing Batch Reactor Activated Sludge Process (SBR), and the Cyclic Activated Sludge System (CASS)
  • "A2O process is" -> The A2O process is ... "SBR process uses" -> the SBR process uses ... "CASS process is adopted" -> the CASS process is adopted
  • "CASS process has the advantages" -> the CASS process has the advantages
  • ... similarly, the authors should use "a" or "the before "PLC" (and "STM32") in many places, depending on the context.

Figures 6, 7 and 9 show very elementary building modules, through possibly a snapshot from their datasheet. Please provide a citation in the caption of the Figure about the datasheet or the source of the Figure.

In chapter 2.3 (and Figure 3) Sensors are described, but Actuators are missed.

Similarly, security is mentioned in ch. 2.3, but no comprehensive material is cited. One of those could be:

Pal Varga, Sandor Plosz, Gabor Soos, Csaba Hegedus, "Security Threats and Issues in Automation IoT", IEEE International Workshop on Factory Communication Systems - a Conference (WFCS 2017), Trondheim, Norway, 2017. DOI: 10.1109/WFCS.2017.7991968

where the authors examine challenges and solutions not only at network-related security but at various levels of IIoT.

The motivation behind using either DSP, ARM or STM32 is very "amateur". This should be improved a lot. (The choice for STM32 to be used in this task comes from its provided functionalities. Maybe a requirement specification would help to make this choice trivial.)

Chapter 5.1: The Title is about the WeChat applet (software system), but it is not clear from the Figure nor the text, where and how exactly WeChat resides in the architecture.

The inclusion of a Message Sequence Chart (with basic communication description among the elements as the text provides it) should improve the paper.

Later, in chapter 5.3.4 the Design of Safety Certification System" is described, however, it is not clear what this has to do with safety - the description and the Figure really refer to security-related attacks. The mitigation seems very basic, though.

Chapter 6 is meant to describe System Tests, but this currently looks like a very basic "user instruction" on how to initiate system tests from the platform, and through the WeChat applet. Test results are really missing here. Even a few should be shown as examples.

The conclusion is very short and misses concluding remarks - at the moment it looks like a very brief summary.

Otherwise, the paper was really interesting to read.

Round 2

Reviewer 2 Report

Authors made a great job in explaining the points that were unclear from the first version of the paper. The data flow diagram is a very good addition, as well as the description of the test procedure.

I would just recommend a quick review of English grammar.